# Learning Robust Rule Representations for Abstract Reasoning via Internal Inferences

**Wenbo Zhang**[1,2,3]  **Likai Tang**[2,3]  **Site Mo**[4]  **Sen Song**[2,3]  **Xianggen Liu**[1,*]

[1]College of Computer Science, Sichuan University
[2]Department of Biomedical Engineering, Tsinghua University
[3]Laboratory of Brain and Intelligence, Tsinghua University
[4]College of Electrical Engineering, Sichuan University
`{zhangwb19, tanglk20}@mails.tsinghua.edu.cn`
`{mosite, xianggenliu}@scu.edu.cn, songsen@tsinghua.edu.cn`

## Abstract

Abstract reasoning, as one of the hallmarks of human intelligence, involves collecting information, identifying abstract rules, and applying the rules to solve new problems. Although neural networks have achieved human-level performances in several tasks, the abstract reasoning techniques still far lag behind due to the complexity of learning and applying the logic rules, especially in an unsupervised manner. In this work, we propose a novel framework, ARII, that learns rule representations for *Abstract Reasoning via Internal Inferences*. The key idea is to repeatedly apply a rule to different instances in hope of having a comprehensive understanding (i.e., representations) of the rule. Specifically, ARII consists of a rule encoder, a reasoner, and an internal referrer. Based on the representations produced by the rule encoder, the reasoner draws the conclusion while the referrer performs internal inferences to regularize rule representations to be robust and generalizable. We evaluate ARII on two benchmark datasets, including PGM and I-RAVEN. We observe that ARII achieves new state-of-the-art records on the majority of the reasoning tasks, including most of the generalization tests in PGM. Our codes are available at `https://github.com/Zhangwenbo0324/ARII`.

## 1 Introduction

Abstract reasoning, the ability to extract patterns and rules from concrete instances and apply them to solve new problems, is one of the hallmarks of human intelligence. It is a critical topic to endow neural networks with the capacity of abstract reasoning on the road from artificial intelligence (AI) to human-like intelligence. As "thinking in pictures" is one of the most effortless and natural ways for humans to perform inference, Raven's Progressive Matrices (RPM) test [1] is a widely accepted task to evaluate AI's reasoning ability. As shown in Fig 1, an RPM problem contains an incomplete $3 \times 3$ matrix where the bottom-right entry is missing. The subjects are aware that the rows in the matrix implicitly share identical rules and are asked to pick the suitable answer from the choices that would best complete the missing entry. Considering that the RPM test is shown to be highly correlated with real intelligence [2] in cognitive and psychological science, AI community recently presents a huge interest in this task to explore the model's capacity of abstract reasoning [3, 4, 5, 6, 7].

In the early years, computational models for RPM usually assume access to the symbolic representations of images and thus solve RPM with heuristics. Recently, various neural networks are proposed

---

*Xianggen Liu is the corresponding author.

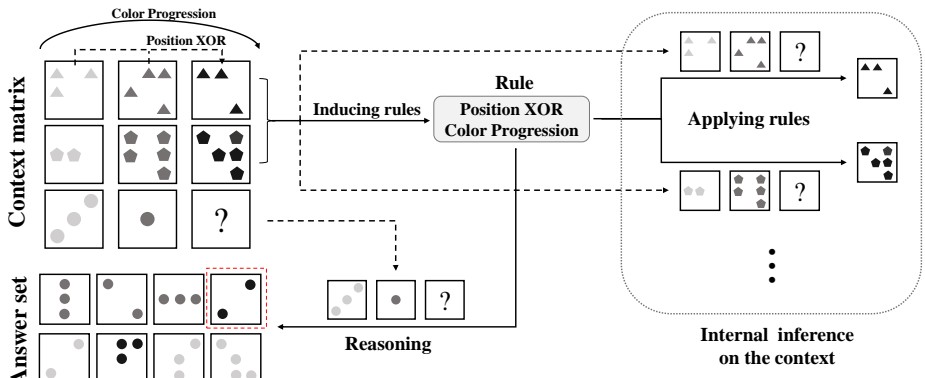

Figure 1: An example of RPM problem and the general solution of ARII. ARII repeatedly applies the rule in internal inference in hope of a deep "understanding" of the rule.

to accomplish abstract reasoning with the help of abundant reasoning data (such as Procedurally Generated Matrices dataset (PGM) [5], Relational and Analogical Visual rEasoNing dataset (RAVEN) [7], and Impartial-RAVEN dataset (I-RAVEN) [8]). These models mainly treat the RPM problem as a classification task based on the rules hidden in rows or columns. For example, Contrastive Perceptual Inference Network (CoPINet) [9] utilizes contrast effect to improve the feature extraction capacity [7]. Scattering Composition Learner (SCL) designs compositions of neural networks according to the specific characteristics in RPM (e.g., the number of the attributes) [10].

Although existing models have achieved impressive performance on abstract reasoning, the critical capacity, i.e., generalization to different environments remains unsolved [5, 11, 12, 13, 14]. In particular, Małkiński [15] reports that, the performances of the current models would significantly decrease even if the test images changes slightly [5, 11, 12, 13]. For example, one of the generalization regimes (i.e., an evaluation task) in the PGM dataset is extrapolation, where the model is trained on the images with color or size restricted to the lower half of the value set and tested on images taking these values from the upper half. Multi-Layer Relation Network (MLRN) [13] achieves nearly perfect accuracy in the normal test of the PGM dataset, but fails completely in the extrapolation regime, yielding only slightly above chance. This reveals that these models are just overfitted to the specific data rather than mastering the reasoning essence, also reflected by low generalization to distracting features [16]. This paper provides a novel framework that ameliorates the reasoning generalizability by learning robust rule representations. To achieve it, we make two assumptions of the rule representation for abstract reasoning. On the one hand, we assume the space of the abstract rules is finite and usually not large. Thus, we propose a rule encoder that produces discrete rule representations by vector quantisation [17]. On the other hand, we assume the rule representations should be invariant to specific instances or distracting features. In human reasoning, people usually apply an abstract rule to different instances several times to deepen the understanding of the rule. We mimic this cognitive process by introducing an internal inference process based on the learned rule representations. Specifically, we randomly mask one panel of the first two rows and ask the referrer to infer the masked panel based on the rule representation. We perform the internal inference multiple times to make the rule representation invariant to specific instances. Based on these instance-invariant representations, conclusions can be made according to the specific context images. Therefore, we call our model ARII, the abbreviation of "abstract reasoning via internal inferences."

We first conduct comprehensive empirical experiments on the I-RAVEN [8] datasets. Compared with the current state-of-the-art models (e.g., SCL [10] and CoPINet [9]), experiments show that ARII achieves new records on four of the seven single training tasks in the I-RAVEN datasets. As the generalization tests can better reflect the real reasoning ability of a model, we also evaluate ARII on the seven generalization regimes of the PGM dataset and observe that ARII outperforms the other competing methods in five regimes. To summarize, our contributions include:

- We propose a novel framework that learns rule representations for abstract reasoning by internal inferences. In particular, we introduce a rule encoder to produce discrete rule representations by vector quantisation. We design an internal inference process to learn an instance-invariant and robust rule representation.

- Our model achieves superior results to state-of-the-art methods on the majority of tasks (four out of seven) on I-RAVEN and better generalization performance on most PGM regimes.

- Visualization results on the rule representations indicate that the learned representations can be automatically clustered with respect to the rule categories. Deep analysis shows that we can identify some features in the discrete representations for a specific reasoning rule, validating that the representations are interpretable and instance-invariant.

## 2 Related Work

RPM problems are firstly investigated in the cognitive science community to better understand intelligence, and many computational models are proposed to address this test since 1990s. For example, Carpenter et al. [2] develop a production system that takes hand-coded textual descriptions of problems as input and predicts the answer. Overall, these cognitive models mainly solve specific problem sets[2] with only a limited number of hand-craft instances in cognitive science [19, 20, 21].

Recently, RPM problems have gained a lot of attention in the AI field to explore abstract reasoning capability. The PGM [5] together with the RAVEN dataset [7] expands the size of RPM instances through automatic generation algorithms, serving as the benchmark datasets for deep learning network to study abstract reasoning. However, Hu et al. [8] found severe defects (a short cut for predictions) existing in the RAVEN dataset and created an unbiased version called I-RAVEN dataset to solve it. Therefore, we evaluate our method on the PGM and I-RAVEN rather than RAVEN.

Based on the above datasets, various end-to-end approaches have been proposed to study the abstract reasoning ability. The typical architectures in computer vision, that simply use CNN as visual feature extractor followed by MLP [3] or LSTM [5, 22] to process the features from all the image panels, are shown to lack the reasoning ability and struggle in the RPM test. To improve the reasoning capacity, Relation Network [23] is equipped and extended in many models, including Wild Relation Network [5] and Multi-scale Relation Network [24].

Although the above models obtain elegant performance on RAVEN and I-RAVEN, they still fail to achieve satisfactory generalization results on PGM, reflecting the overfitting issue of these reasoning models. Auxiliary labels [25, 11, 16] are therefore utilized to develop representations that are amenable to generalization by informing the meta-targets (i.e., labels which encode relevant relations and attributes). However, it requires additional prior information, which is not a general way to address the reasoning problem.

Our work is closely related to several advanced abstract reasoners, such as the SCL [10] and MXGNet [11], a multi-layer multiplex graph neural net. In particular, ARII uses the splitting and grouping embeddings and row-wise embeddings in SCL and MXGNet, respectively. Our work is also connected with neuro-symbolic methods for logical reasoning [26, 27, 28, 29, 30, 31, 32]. For instance, Logic Tensor Networks integrates neural networks with first-order fuzzy logic to perform reasoning with logical formulas. Different from these studies, ARII explicitly maintains a discrete rule representation for abstract reasoning and introduces a novel internal inference process to enforce the rule representation to be robust. The internal inference can be viewed as one type of self-supervised learning algorithm for the rule representation learning, where we mask parts of the context matrix and require the model to reconstruct it. But the traditional self-supervised learning models are usually trained first and finetuned for the downstream tasks afterwards [33, 34, 35]. Different from them, we explore an interesting strategy that performs the internal inference and the supervised learning simultaneously.

## 3 Methodology

### 3.1 Problem formulation

The RPM test is conducted on a $3\times3$ matrix where the rows comply with an identical rule. Subjects are presented with an incomplete matrix where an entry on the third row is missing, and are asked to select a candidate answer from the choice set to complete the matrix.

---

[2]Standard Progressive Matrices [18] contains 60 problems

Formally speaking, a RPM problem contains 16 image panels $X = \{x_i\}_{i=1}^{16}$ divided into 8 context images $X_c = \{x_i\}_{i=1}^{8}$ and 8 candidate answer images $X_a = \{x_i\}_{i=9}^{16}$. The 8 context images are placed in a $3 \times 3$ matrix with the last element blank. For each particular data point, the three rows in the matrix implicitly share the identical abstract rule. Given the $X$, the machine is asked to select a candidate answer image from $X_a$ to best complete the matrix (i.e. adhering to the underlying abstract rule in context images). The correct answer is denoted by $x^*$, where $x^* \in X_a$.

## 3.2 The ARII model

ARII consists of three modules: the rule encoder, the reasoner, and the internal inferrer. The rule encoder induces rule representations from the context matrix. The inferrer performs internal inferences to regularize the rule representations to be robust and instance invariant. The reasoner draws the reasoning conclusion based on the rule representations. The rest of this section elaborates on these components and the training process.

## 3.3 Rule encoder

To tackle this reasoning problem, an ideal reasoning process is to figure out the underlying rule based on the given context images. To achieve this, we aim to learn the rule representations $r$ using an effective encoder. In principle, the rule can be extracted from particular row of images. But the above extraction has the potential to make mistakes since a single row of images may satisfy multiple rules. Therefore, to guarantee the adequate information for rule extraction, we propose to encode the first two rows of images to represent the underlying rule, given by

$$r_{1\&2} = \text{RuleEnc}(\{x_i\}_{i=1}^{6}), \tag{1}$$

where $r_{1\&2}$ stands for the representation of the rule underlying the first two rows in the given matrix. Rule-Enc stands for the rule encoder in the ARII model. In the rest of this section, we first obtain the image encodings and then build rule representations based on the encodings of the involved images.

**Image encodings.** As each row contains three separated images, we adopt convolutional neural network (CNN) blocks to get the feature map ($f$) for each image separately.

$$f_i = \text{CNN}(x_i), \quad i = 1, 2\ldots, 6. \tag{2}$$

Once we get the feature map, we further fuse the features of different positions in the feature map. Intuitively, the rule of RPM is highly correlated with visuospatial ability [36] and usually lies in the same position of different panels. Therefore, we first divide the feature map into multiple groups; each group can be considered to contain information from a certain position. Then we apply a fully connected layer (FC) to each group to further extract group features, and concatenate the group features into output features, computed by

$$z_i = \text{ImageEnc}(x_i) = ||_k^{K_z}[\text{FC}(f_i^k)], \quad i = 1, 2\ldots, 6, \tag{3}$$

where $||$ stands for the concatenation operation of the vectors. $f_i^k$ is $k$-th group features after splitting original $f_i$ into $K_z$ groups. Similar to the scattering transformation in SCL [10], the group features $\{f_i^k\}$ are divided evenly from the feature map $f_i$. Finally, the $K_z$ new groups of the features are concatenated (denoted by $||_k^{K_z}$) to obtain the $i$-th image encodings $z_i$.

**Continuous rule representations.** Next, the rule encoder fuses the image encodings and extracts the rule representations. Similar to the processing of images, we extract the relation corresponding to the different positions of the six image encodings as Fig 2. Specifically, the encodings of six images are first divided into $K_r$ groups independently. The groups at the same location in six image encodings are combined together and fed into a fully connected layer to further extract the relation feature. Finally, the multiple relation features are merged to form rule features as follows:

$$r_c = ||_k^{K_r}[\text{FC}(||_{i=1}^{6} z_i^k)], \tag{4}$$

where $z_i^k$ is the $k$-th group of the $i$-th image encodings $z_i$. $||_{i=1}^{6} z_i^k$ means six $k$-th group features from encodings are concatenated, and fed into a fully connected layer (FC) to get the $k$-th relation features. We further merge the $K_r$ relation features and obtain the continuous rule representations $r_c$.

**Discrete rule representations.** Note that the above rule representations $r_c$ are continuous and dependent on the specific instances. In this work, we assume the space of the abstract rules is finite

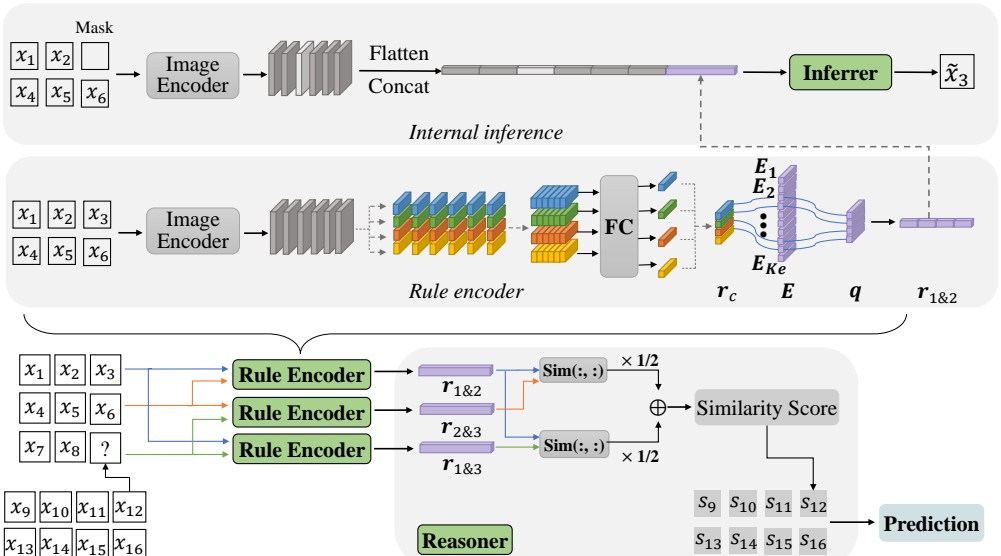

Figure 2: The architecture of ARII. ARII comprises three modules, namely, the rule encoder, the reasoner and the internal inferrer.

and usually not large. Based on this assumption, the discrete and finite representations are more suitable to describe abstract rules.

Therefore, we propose to discretize the latent representations of the rules via vector quantisation (VQ) [17]. The general idea of VQ is to use a finite number of vectors (denoted as codebook) to build the rule space. It outputs the closest vector from the codebook for the given input (Appendix Fig 1). In particular, the rule encoder maintains a discrete codebook $E \in \mathbb{R}^{K_e \times D}$. $K_e$ is the number of the discrete code, and $D$ is the dimensionality of each discrete code $E_k$. We measure the L2 distance between the continuous representation $\boldsymbol{r}_c$ and code vectors in the codebook. For each vector $\boldsymbol{r}_c^l$ in $\boldsymbol{r}_c \in \mathbb{R}^{K_r \times D}$, where $K_r$ is the number of the continuous vector and $D$ is the dimensionality of each continuous vector $\boldsymbol{r}_c^l$, the code vector that yields the minimum distance is taken to obtain the discrete rule representation $\boldsymbol{q}_l$. The detailed computations are

$$\boldsymbol{q}_l = E_k, \quad \text{where} \quad k = \underset{j \in \{1, \dots, K_e\}}{\arg \min} \| \boldsymbol{r}_c^l - E_j \|_2, \tag{5}$$

where $\boldsymbol{q}_l$ is the closest quantized vector for the continuous vector $\boldsymbol{r}_c^l$. Via vector quantisation, we can find the closest code for the given continuous rule representations, i.e.,

$$\boldsymbol{r}_{1\&2} = \text{RuleEnc}(\{x_i\}_{i=1}^6) = \|_{l=1}^{K_r}[\boldsymbol{q}_l]. \tag{6}$$

To make the above vector quantisation work, we need to train both the neural networks and the codebook in the rule encoder towards the minimum of the distance between the continuous and discrete representations. The objective function of vector quantisation for a particular data point is

$$\mathcal{J}^{\text{vq}}(X) = \|\text{sg}(\boldsymbol{r}_c) - \boldsymbol{r}_{1\&2}\|_2^2 + \|\boldsymbol{r}_c - \text{sg}(\boldsymbol{r}_{1\&2})\|_2^2, \tag{7}$$

where $\text{sg}(\cdot)$ is stop-gradient operation during computation. In this way, we can derive the discrete rule representations, which is expected to be interpretable and instance-invariant.

### 3.4 Reasoner

The reasoner is to draw conclusions (i.e., the selection of answer images) based on the extracted rule and the specific context (i.e., the images on the third row). That is, the reasoner applies the extracted rule to the third row and figures out which candidate answer could best complete the missing entry.

According to this standard, we therefore extract the rule that the third row implicates. Since we have 8 candidate answer images (i.e., $X_a$), there are 8 rules for the third row with the different answer images in the missing entry. However, the rule encoder is designed to take two rows of images as inputs and yield the rule representations. Therefore, we combine the third row with the first row to

derive the rule that these two rows implicitly share, computed by

$$r_{1\&3}^j = \text{RuleEnc}(\{x_i | i = 1, 2, 3, 7, 8, j\}), \quad j = 9, \ldots, 16 \tag{8}$$

where $r_{1\&3}^j$ denotes the rule representations of the rule underlying the first and the third row when the third row is completed with the $j$-th answer image. Similarly, we can obtain the rule representations of the rule underlying the second and third row, given by

$$r_{2\&3}^j = \text{RuleEnc}(\{x_i | i = 4, 5, 6, 7, 8, j\}), \quad j = 9, \ldots, 16 \tag{9}$$

Based on the derived rule representations for the third row, the reasoner next measures the similarity between these rule representations and the reference rule (i.e., $r_{1\&2}$) and selects the most similar one as the prediction. In particular, the reasoner measures the similarity of two rules by calculating the inner product of the corresponding representations. For example, the similarity between the rule $r_{1\&3}$ and the reference rule is calculated by

$$\text{sim}(r_{1\&3}^j, r_{1\&2}) = \frac{e^{r_{1\&3}^j \cdot r_{1\&2}}}{\sum_{j=9}^{16} e^{r_{1\&3}^j \cdot r_{1\&2}}}, \quad j = 9, \ldots, 16, \tag{10}$$

where $\text{sim}(\cdot)$ is the similarity function of two rule representations. Similarly, the reasoner derives the similarity between the rule $r_{2\&3}$ and the reference rule by

$$\text{sim}(r_{2\&3}^j, r_{1\&2}) = \frac{e^{r_{2\&3}^j \cdot r_{1\&2}}}{\sum_{j=9}^{16} e^{r_{2\&3}^j \cdot r_{1\&2}}}, \quad j = 9, \ldots, 16. \tag{11}$$

Based on the similarities between individual rules, the reasoner then predicts the candidate answer $y$.

$$y = \underset{j \in \{9, \ldots, 16\}}{\arg\max} \frac{1}{2} [\text{sim}(r_{1\&3}^j, r_{1\&2}) + \text{sim}(r_{2\&3}^j, r_{1\&2})]. \tag{12}$$

For the optimization of the reasoner, we adopt the cross entropy loss to guide the reasoner to select the correct answer, given by

$$\mathcal{J}^{\text{re}}(X) = -\frac{1}{2} [\log \text{sim}(r_{1\&3}^*, r_{1\&2}) + \log \text{sim}(r_{2\&3}^*, r_{1\&2})] \tag{13}$$

### 3.5 Internal inferences

Human usually applies an abstract rule to different instances for several times to deepen the understanding of the rule. We suppose that the underlying rules are independent of specific instances and can be reused in different problems that share the same rules. Therefore we mimic this cognitive process by performing internal inferences based on the rule representations. In the internal inference, we introduce another reasoner to address the reasoning task on the two rows of images when the rule representations are available. In this way, our method ARII could also better "understand" the rule and have more robust rule representations. A robust rule representation is insensitive to the sample noises and thus benefits the overall reasoning ability[37].

In particular, the first and second rows are used in internal inference process. We first extract the rule representations $r_{1\&2}$ of these rows and then randomly mask one of the six images $\{x_i\}_{i=1}^6$ by a white blank image. We coin the blank image as $\hat{x}_m$, where $m$ is the index of the masked images. Thus, the masked two rows can be expressed by

$$I_m = \{x_i | i \in \{1, 2, 3, 4, 5, 6\} \backslash m\} \cup \{\hat{x}_m\} \tag{14}$$

where $\{1, 2, 3, 4, 5, 6\} \backslash m$ stands for a set excluding the index $m$ and we insert blank image $\hat{x}_m$ in the original $m$-th position. Next, we introduce another neural network named inferrer that takes the rule representations and the encodings of the images in the masked two rows as inputs.

$$\tilde{x}_m = \text{Inferrer}(\text{ImageEnc}(I_m), r_{1\&2}), \tag{15}$$

where Inferrer is another CNN model. $\tilde{x}_m$ stands for the predicted image for the incomplete rows, and $\text{ImageEnc}(I_m)$ derives the encoding of $I_m$ by Equation 2 and 3. To avoid using additional data in our internal inference, we formulate the above prediction as to the generation process. That is to say, the inferrer aims to generate the masked image $x_m$ at the pixel level.

In addition, we repeat the above internal inferences for 6 times by masking each image to learn more robust rule representations. We adopt mean squared error (MSE) as the objective function to optimize

Table 1: Test accuracy of individual models on I-RAVEN.

| Model | Test Accuracy (%) | | | | | | | |
|---|---|---|---|---|---|---|---|---|
| | Average | Center | 2×2 Grid | 3×3 Grid | L-R | U-D | O-IC | O-IG |
| LSTM [7] | 12.5 | 12.3 | 13.3 | 12.8 | 12.7 | 10.3 | 12.9 | 13.1 |
| WReN [5] | 17.8 | 23.3 | 18.1 | 17.4 | 16.5 | 15.2 | 16.8 | 17.3 |
| CNN+MLP [5] | 12.9 | 12.9 | 13.2 | 12.7 | 11.5 | 13.5 | 12.9 | 13.7 |
| Resnet-18 [7] | 14.5 | 20.8 | 12.9 | 14.3 | 13.2 | 13.4 | 13.8 | 12.9 |
| LEN [16] | 28.4 | 42.5 | 21.1 | 19.9 | 27.6 | 28.1 | 32.9 | 27.0 |
| CoPINet [9] | 38.6 | 50.4 | 30.9 | 28.5 | 40.0 | 40.8 | 42.7 | 36.9 |
| SCL [10] | 85.4 | **99.8** | 72.4 | 64.2 | **99.5** | **99.4** | 98.6 | 64.2 |
| Ours | **91.1** | 98.9 | **88.2** | **78.9** | 97.8 | 90.3 | **98.7** | **85.1** |

the inferrer as well as the rule encoder. In this way, we obtain the overall objective function $\mathcal{J}^{\text{if}}$ by collecting the predictions of these internal inference processes, given by

$$\mathcal{J}^{\text{if}}(X_c) = \frac{1}{6} \sum_{m=1}^{6} ||x_m - \tilde{x}_m)||_2^2 \qquad (16)$$

## 3.6 Optimization of ARII

As introduced in the section on problem formulation, the major goal of our method ARII is to select a candidate answer image for the incomplete matrix as accurately as possible (as shown in Equation 13). In the meanwhile, we assume the rule representations are discrete and can be reused in internal inferences. That is, apart from the major goal, we have two additional objectives, i.e., the objective of the vector quantisation process and the objective of the internal inferences. Therefore, we combine these three objectives together for the optimization of the individual components of ARII, given by

$$\mathcal{J} = \sum_{X \in \mathcal{D}} \lambda_{\text{re}} \mathcal{J}^{\text{re}}(X) + \lambda_{\text{vq}} \mathcal{J}^{\text{vq}}(X) + \lambda_{\text{if}} \mathcal{J}^{\text{if}}(X_c), \qquad (17)$$

where the context images $X_c$ is parts of the given data point $X$ and $\mathcal{D}$ is the dataset. $\lambda_{\text{re}}$, $\lambda_{\text{vq}}$ and $\lambda_{\text{if}}$ are the three hyper-parameters that coordinates the importance of the components in ARII.

# 4 Experiment

## 4.1 Datasets

**PGM.** Each matrix in PGM is governed by abstract rules, which are sampled from a rule set $\mathcal{R} = \{[s, o, a] : s \in \mathcal{S}, o \in \mathcal{O}, a \in \mathcal{A}\}$, where $\mathcal{S}, \mathcal{O}, \mathcal{A}$ are primitive sets of relations, objects and attributes, respectively. $\mathcal{S} = \{\text{progression, XOR, OR, AND, consistent union}\}$, $\mathcal{O} = \{\text{shape, line}\}$, $\mathcal{A} = \{\text{size, type, color, position, number}\}$. There are eight regimes in PGM. The easiest one is called neutral where the training and test set are sampled from the same distribution, and the others are generalization regimes where the training and test data differ in the pre-defined ways. The generalization regimes consists of seven variants: interpolation, extrapolation, held-out triples (H.O.Triples), held-out attribute pairs (H.O.A.P.), held-out pairs of triples (H.O.T.P.), held-out shape-colour (H.O.S-C) and held-out line-type (H.O.L-T). More details can be found in Barrett et al. [5].

**RAVEN and I-RAVEN.** The RAVEN and I-RAVEN datasets are extensions to PGM. They share an identical set of attributes but differ in the following attributes: progression, constant and arithmetic. I-RAVEN is adapted from RAVEN dataset for its biased answer sets which may lead to a shortcut solution that only candidate answers can yield accurate answers [8]. In our experiments, we first evaluate the models' reasoning ability on I-RAVEN since it is easier than PGM.

## 4.2 Competing methods and implementation details

We compare our model with several state-of-the-art models, including CNN + LSTM [5], ResNet-based [38] image classifier, Wild ResNet [5], Wild Relation Network (WReN) [5], Contrastive Perceptual Inference network (CoPINet) [9], Logic Embedding Network (LEN) [16], Scattering Compositional Learner (SCL) [10], Stratified Rule-Aware Network (SRAN) [8], Multi-scale Relation

Table 2: Test accuracy of different models on PGM

| Model | Test Accuracy (%) | | | | | | | |
|---|---|---|---|---|---|---|---|---|
| | Neutral | Interpolation | H.O.A.P. | H.O.T.P. | H.O.Triples | H.O.L-T | H.O.S-C | Extrapolation |
| CNN MLP [5] | 33.0 | - | - | - | - | - | - | - |
| CNN LSTM [5] | 35.8 | - | - | - | - | - | - | - |
| ResNet-50 [5] | 42.0 | - | - | - | - | - | - | - |
| Wild-ResNet [5] | 48.0 | - | - | - | - | - | - | - |
| CoPINet [9] | 56.4 | - | - | - | - | - | - | - |
| WReN $\beta = 0$ [5] | 62.6 | 64.4 | 27.2 | 41.9 | 19.0 | 14.4 | 12.5 | 17.2 |
| VAE-WReN [39] | 64.2 | - | 36.8 | 43.6 | 24.6 | - | - | - |
| MXGNet $\beta = 0$ [11] | 66.7 | 65.4 | 33.6 | 43.3 | 19.9 | 16.7 | 16.6 | 18.9 |
| LEN $\beta = 0$ [16] | 68.1 | - | - | - | - | - | - | - |
| DCNet [12] | 68.6 | 59.7 | - | - | - | - | - | 17.8 |
| T-LEN $\beta = 0$ [16] | 70.3 | - | - | - | - | - | - | - |
| SRAN [8] | 71.3 | - | - | - | - | - | - | - |
| Rel-Base [40] | 85.5 | - | - | - | - | - | - | 22.1 |
| SCL [10] | 88.9 | - | - | - | - | - | - | - |
| MRNet [24] | 93.4 | 68.1 | 38.4 | 55.3 | 25.9 | **30.1** | **16.9** | 19.2 |
| MLRN [13] | **98.0** | 57.8 | - | - | - | - | - | 14.9 |
| Ours | 88.0 | **72.0** | **50.0** | **64.1** | **32.1** | 16.0 | 12.7 | **29.0** |

Table 3: Ablation study on two regimes in PGM

| Regime | w/o discretization | w/o internal inference | w/o both | ARII |
|---|---|---|---|---|
| H.O.A.P | 47.5 | 32.0 | 36.6 | **50.0** |
| Interpolation | 69.5 | 54.2 | 51.1 | **72.0** |

Network (MRNet) [24], and Multi-Layer Relation Network (MLRN) [13]. Note that our model is trained end to end and needs solely the ground truth answer label without auxiliary information (i.e., rule labels). We report the result of the baseline models trained without auxiliary labels for fair comparison. Please refer to Appendix A for more implementation details of ARII.

## 4.3 Results

Table 1 shows the test performance of individual models on I-RAVEN dataset. We observe that the typical architectures in computer vision, that simply use CNN as visual feature extractor followed by MLP or LSTM, fail to yield satisfactory reasoning results. More advanced models for abstract reasoning such as LEN and SCL significantly improve the performance. In particular, the previous state-of-the-art model SCL achieves nearly 100% accuracy on the configurations of Center, L-R, U-D, and O-IC, but SCL could not make similar success on the other configurations. We conjecture that the reasoning complexity of these four configurations is relatively low and easy. As for our method, ARII performs similarly to SCL on the above four configurations but largely outperforms it on the other three more difficult configurations, leading to a notably higher average score on I-RAVEN.

In addition, Table 2 presents the performance on the PGM dataset. For the neutral regime, our model achieves a decent result that is better than majority of baseline models, although it does not outperform state-of-the-art models such as MRNet and MLRN. Note that MLRN achieves near perfect accuracy in the neutral regime due to the sophisticated design which could not be applied to other problems. More importantly, the neutral regime is not our main focus of metric, since it could hardly indicate the reasoning ability and generalization towards real intelligence.

PGM also allows us to investigate the generalization capacity of the models under various regimes. We observe from Table 2 that although MLRN achieves superior performance on neutral, it is prone to overfitting and performs poorly on the interpolation and extrapolation regimes, clearly indicating MLRN is overfitted to the neutral setting. On the contrary, the proposed internal inference in ARII serves as the regularities of the rule representations, making the rule representations robust and instance-invariant. As a result, our model outperforms all other methods on five of the seven generalization regimes in PGM dataset. In particular, the accuracy of ARII increases by a sizable margin over the baselines in the H.O.A.P. and H.O.T.P. regimes (around 10%), further validating the superior generalization ability of abstract reasoning.

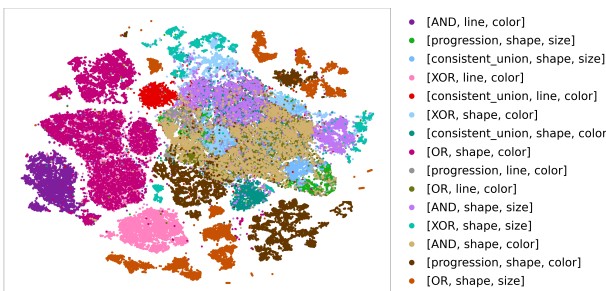

Figure 3: Visualization of rule embeddings extracted from rule encoder. Each point stands for the rule representation from a data point in the test set.

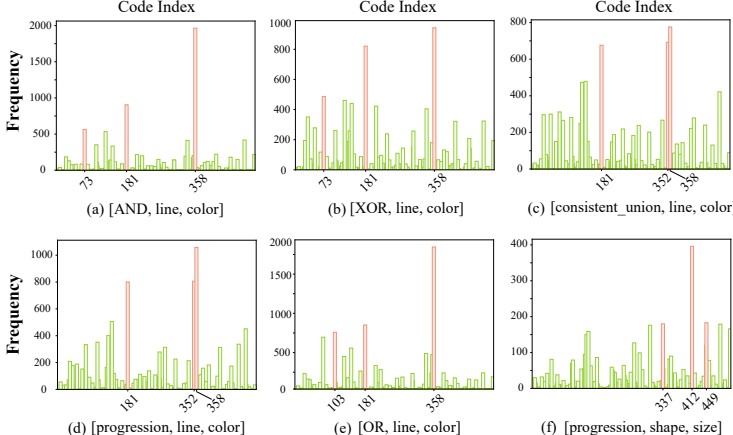

Figure 4: The selected frequencies of codes in rule representations. We take the rules related to the line object and the color attribute as examples. Similar results for other rules are in Appendix Fig 2.

## 4.4 Ablation study

We analyze ARII in more detail to validate the effectiveness of the proposed components on two regimes in the PGM dataset (Table 3). In the H.O.A.P. regime, the performance of ARII without discretization decreases by 2.5%, which shows that discrete representation plays its role in generalization of abstract reasoning. The performance is also largely declined by 18% when the internal inference is removed, suggesting that the internal inference is vital to learn a more robust rule representation for abstract reasoning. Similar results can be found in the interpolation regime.

We further conduct ablation studies to investigate the impact of generation in internal inference, the combinatorial benefit of $r_{1\&3}$ and $r_{2\&3}$ in the reasoner, and effectiveness of the internal inference to other models. Please see Appendix B-D for more discussion.

## 4.5 Visualization of rule representations

Then, we try to validate whether ARII has learned the instance-invariant representations. We first obtain all the rule representations in the test set of the interpolation regime of PGM, and use the representations derived from the problems which contain only a single rule (i.e., a triple $\{[s, o, a]\}$). The rule representations are projected to 2D space by t-distributed stochastic neighbor embedding (t-SNE) [41] in Fig 3. We observe that the rule representations are clustered according to individual rule triples. Note that these clusters for each rule are formed without supervision of the rule labels. And the representations are extracted from the generalization test set which has a different distribution from the training set. The clustered representations show that these rule representations are universal and independent to the specific instances, even to the out-of-distribution brand new instances.

Taking one step further, we investigate whether the discrete rule representation is interpretable. In particular, we want to see the composition of the representations for each specific rule. Here, we take

the rules related to the line object and the color attribute (totally five rules, i.e., $[s \in \mathcal{S},$ line, color], denoted by the $[\cdot,$line, color] rules) as an example. We first collect the rule representations from all the instances governed by these rules, which are sampled from the codebook $E$. Then we calculate the selected frequencies of the codes at the 30-th position in the rule representations and visualize their distributions. Fig 4 (a-e) shows that all the five rules share the same codes $E_{358}$ and $E_{181}$ in their top-3 codes, indicating that the codes $E_{358}$ and $E_{181}$ are very relevant to the $[\cdot,$line, color] rules. In addition, these codes are also specific to these rules, since they rarely appear in the dissimilar rules such as the rule [progression, shape, size] (Fig 4 (f)). These results reveal that the rule representations learned by ARII are moderately interpretable. Besides, we conduct a classification experiment to see whether the rules can be classified from the rule representations. The classification accuracy based on the rule representations from ARII is 72.1%, largely higher than a baseline model SRAN [8] that yields 59.4%. This indicates the superiority of ARII which can capture the underlying rules and learn more robust rule representations.

## 5    Limitations

Abstract reasoning is the ability that consciously applies logic from premises to conclusion [42, 43]. For neural networks, they use different layers of the distributed neurons to implicitly represent the premises, the logic and the temporal results, and formulate the final conclusion-making process as the classification or generation task. Therefore, neural networks lack in explaining how they perform induction and reasoning. Although our method takes one step further to build interpretable rule representations, it is still far from the goal of a fully interpretable, robust, and accurate reasoner for general reasoning. Another limitation of this work is the image encoder, which relies on the rigid spatial structure and is difficult for learning new rules in other abstract reasoning problems. The general object detector pre-trained on natural images is a potential solution.

## 6    Conclusion

In this work, we propose a novel architecture ARII that learns robust rule representations for abstract reasoning via internal inferences. Experiments on benchmark datasets (I-RAVEN and PGM) show that ARII outperforms previous state-of-the-art methods without using auxiliary annotations on the majority of the reasoning tasks. Moreover, the rule representations of ARII present meaningful and interpretable characteristics, facilitating the exploration of the composition of rules in future work.

## Acknowledgements

This work was supported by the National Natural Science Foundation of China under Grant 61836004 and 62206192, and a grant from Institute Guo Qiang, Tsinghua University.

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
