# Appendix: Learning Robust Rule Representations for Abstract Reasoning via Internal Inferences

## A   Implementation details

In ARII, the convolutional block used in rule encoder is 4-layer CNN with kernel size 3 and channels of $\{16, 16, 32, 32\}$. The group number $K_z$ for extracting image encodings is 10. The number of codes $K_r$ in the rule representation is set to 80. The dimensions of the codebook $E$ are $K_e = 512$ and $D = 5$. The inferrer is another CNN block (3-layer CNN with kernel size 3 and channels $\{64, 32, 1\}$) with upsampling techniques (before each of the first two layers with scale factor 2).

The batch size is set to 32 and learning rate 0.001. The weights in the objective function $\{\lambda_{\mathrm{re}}, \lambda_{\mathrm{vq}}, \lambda_{\mathrm{if}}\}$ are $\{1, 1, 1\}$ for I-RAVEN dataset and $\{1, 1, 1\}$ or $\{0.1, 0.1, 0.8\}$ for PGM dataset. We use the Adam[1] optimizer and apply weight decay from $\{0, 0.1, 0.01\}$ in individual training tasks. We train the model for 10 epochs for each tasks and report the performances on the test sets. Our experiments are conducted on Nvidia 3090 GPUs with 24 GB RAM. For model training, we use single training setting [2], where the model is trained and tested on each configuration in I-RAVEN or regime in PGM separately.

## B   Design choices of the internal inference

We conduct an ablation study on the internal inferrer module to investigate the role played by the generative process. In particular, we replace generative process with classification, denoted by "ARII (classification)". In the classification task, the input is the two rows of images where one is masked with blank (i.e., $I_m$ in Equation 14, the same as the original ARII). The classification choices are the existing context images in the instance and the correct answer is the particular image we mask out. The inferrer reuses the reasoner module to make the classification decisions. Apart from the above changes, the other settings are the same with the original ARII.

Appendix Table 1 reports the results of the classification variant of ARII on two regimes of the PGM dataset. We observed that the classification variant yields lower performance than the generative variant on the interpolation regime. These results indicate that the classification process could also serve as a reasonable task in the internal inference module but the generative process is better. We conjecture that the reason for the poor performance is that the rule representation yielded by the correct answer (i.e., the masked image) is actually the same as the reference rule representation extracted from the first two rows. The similarity score of the two will always be the highest one. The reasoner module will always choose the masked image, which does not affect the reference rule representations and is less useful for the learning of rule representations. This ablation study further demonstrates that our internal inference process plays a critical role in visual reasoning.

## C   Combinatorial benefit of rule representations in the ARII reasoner

To investigate the combinatorial benefit of using $r_{1\&2}$ with $r_{1\&3}$ and $r_{2\&3}$, we remove the information of $r_{2\&3}$ from inputs of the reasoner and report the performance in Appendix Table 2. We observe that the reasoning performance decreases slightly compared to the original model. This result indicates that combination of using $r_{1\&2}$ with $r_{1\&3}$ and $r_{2\&3}$ is beneficial to the reasoner but it is not the main

reason for the performance gain of ARII. Instead, Table 3 (the third column) in main text presents that the internal inference could significantly improve the robustness of the reasoning.

## D  Effectiveness of the internal inference to other models

We are curious about whether the internal inference process is plugged and played and we apply the internal inference module to other models. However, since the internal inference takes the rule representation as input, most of the previous methods do not satisfy the requirement of the internal inference. That is, most previous methods do not explicitly have a rule representation during the reasoning process. We only find a suitable one, SRAN [3]. We adapt its released code and impose the internal inference module into its framework. The reasoning performance is listed in Appendix Table 3.

Experimental results show that SRAN with internal inference yields better performance than the original SRAN. We notice that the internal inference process does not produce significant performance improvement. We conjecture the reason is that the rule encoder of SRAN can access full information of the context (our rule encoder is restricted to two rows and outputs symbolic features). The regularization of the internal inference can only play a moderate role. However, the consistently better results on the two regimes of PGM indicate that the internal inference is plugged and played for the other models.

## E  Appendix Tables

Table 1: Ablation study on generation and classification in internal inference.

| Regime | ARII (generation) | ARII (classification) |
|---|---|---|
| Interpolation | 72.0 | 62.4 |
| H.O.A.P | 50.0 | 37.7 |

Table 2: Ablation study on reasoner with or without $r_{2\&3}$.

| Regime | ARII ($r_{1\&3}$ & $r_{2\&3}$) | ARII (w/o $r_{2\&3}$) |
|---|---|---|
| Interpolation | 72.0 | 69.5 |

Table 3: Effectiveness of internal inference to SRAN.

| Regime | SRAN | SRAN (with internal inference) |
|---|---|---|
| Interpolation | 56.4 | 58.9 |
| H.O.A.P | 33.5 | 34.5 |

## F  Appendix Figures

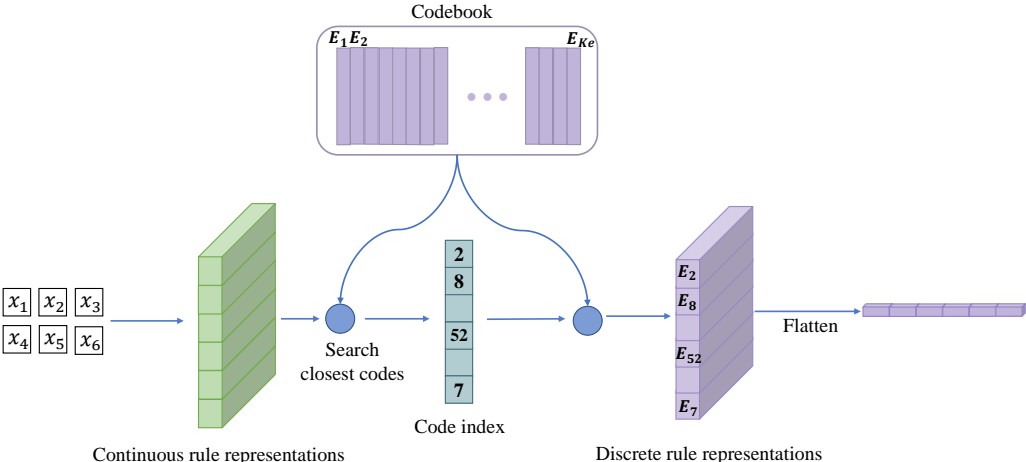

Figure 1: The architecture of vector quantization.

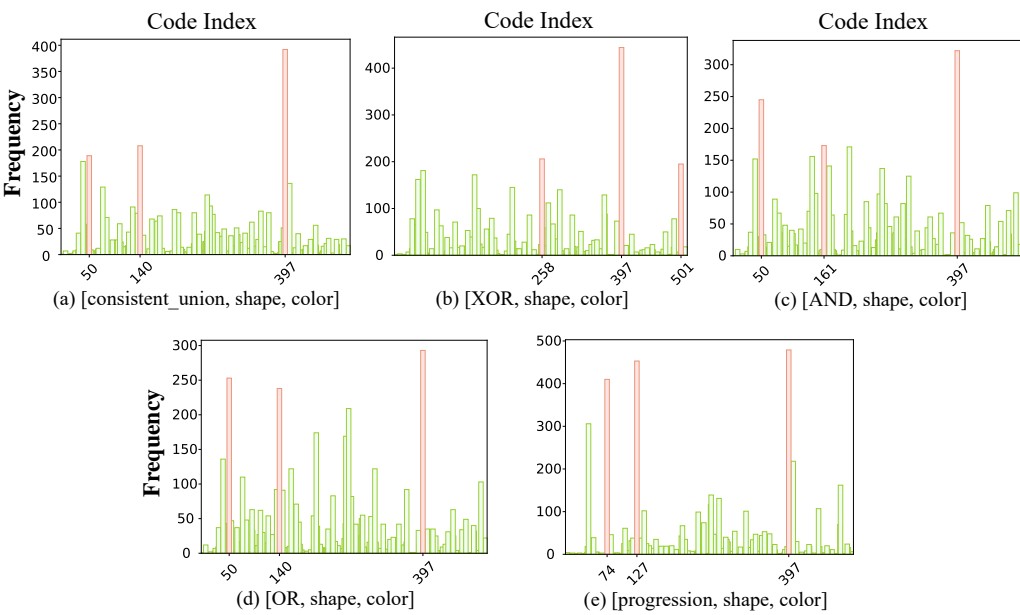

(a) [consistent_union, shape, color]

(b) [XOR, shape, color]

(c) [AND, shape, color]

(d) [OR, shape, color]

(e) [progression, shape, color]

Figure 2: The selected frequency of code in the representations of the rules [·,shape, color].