# OpenReview forum: "Learning Robust Rule Representations for Abstract Reasoning via Internal Inferences"
_NeurIPS.cc/2022/Conference — NeurIPS 2022 Accept_

### Official Review · Reviewer_xDma · 2022-07-10

**Rating:** 5
**Confidence:** 5
**Soundness:** 3 good
**Presentation:** 3 good
**Contribution:** 3 good

**Summary:**

A model called ARII is proposed in this work to solve abstract reasoning problems. The ARII model is composed of a rule encoder, a reasoner, and an internal inferrer. The rule encoder adopts a vector quantization method and turns continuous rule representation from the first two roles into a discrete token in the codebook. The reasoner uses a similarity-based measure to select the answer: the candidate which, when filled in, generates rule features most similar to the first two rows will be treated as the answer. In addition, the internal inferrer reconstructs a masked panel like in BERT. In experiments, the authors conduct evaluation on I-RAVEN and PGM. It's found that the model achieves best performance on I-RAVEN and improves on the generalization regimes on PGM.

**Questions:**

1. It would be nice to see results of existing models trained with the masked modeling loss.
2. How the reasoning capability emerges.

**Limitations:**

No. Limitations are not adequately addressed, though there is not ethical issues.

**Strengths And Weaknesses:**

The paper is well written with clear explanation and a streamlined flow.

There are a few things that are done right which I believe lead to the performance boost in this work. (1) image features are extracted from a group level: considering the grid-like layout in I-RAVEN and PGM, the position-based grouping could help better capture the statistics in the images. (2) Quantization in rules: as there are only a finite number of rules, quantization seems a reasonable choice, which is unfortunately missing in existing works. (3) BERT-like masked reconstruction. This might be the part that contributes to the generalization improvement as hinted in existing language modeling works.

I'm wondering, as the masked reconstruction part can be plugged and played, whether performance in generalization could be consistently improved in different models. It would be nice to see some results regarding this point.

Also I wonder how reasoning is conducted. It occurs to me that the work still applies statistical learning but lacking in explaining how a model performs induction and reasoning. Some statistics may be captured using the method, but as shown in generalization results (which are indeed improved), it is still far below fair expectation. I would be interested in knowing how the authors view the perspective: does reasoning emerge from statistical learning and if not what is your opinion.

Finally, I notice a few neuro-symbolic methods [1, 2] are missing, which when added should better complete the full picture of the area.

[1] Zhang, Chi, et al. "Abstract spatial-temporal reasoning via probabilistic abduction and execution." Proceedings of the IEEE/CVF Conference on Computer Vision and Pattern Recognition. 2021.
[2] Zhang, Chi, et al. "Learning algebraic representation for systematic generalization in abstract reasoning." arXiv preprint arXiv:2111.12990 (2021).

---

> ### Author Response · Authors · 2022-08-02
> **Response to Reviewer xDma**
>
> We thank the reviewer for the thoughtful comments and will response to the comments point by point.
>
> **Q1: It would be nice to see the results of existing models trained with the masked modeling loss.**
>
> **A1:** We are also curious about whether the internal inference module (i.e., the masked reconstruction part) is plugged and played. Thus we follow the reviewer’s suggestion to apply the internal inference module to other models. However, since the internal inference takes the rule representation as input, most of the previous methods do not satisfy the requirement of the internal inference. That is, most previous methods do not explicitly have a rule representation during the reasoning process. We only find one that is suitable, that is, SRAN [1]. We adapt its released code and impose the internal inference module into its framework. The reasoning performance is listed in the below table.
>
> | Regime        | SRAN | SRAN (with internal inference) |
> |---------------|------|--------------------------------|
> | Interpolation | 56.4 | 58.9                           |
> | H.O.A.P       | 25.0 | 26.4                           |
>
> Experimental results show that SRAN with internal inference yields better performance than the original SRAN. We notice that the internal inference process does not produce significant performance improvement. We conjecture the reason is that the rule encoder of SRAN can access full information of the context (our rule encoder is restricted to two rows and outputs symbolic features). The regularization of the internal inference can only play a moderate role. However, the consistently better results on the two regimes of PGM indicate that the internal inference is plug and play for the other models. Thanks for recommending this ablation study. This would significantly improve our paper.
>
> **Q2: How the reasoning capability emerges.**
>
> **A2:** One of the widely used definitions of reasoning is “reasoning is the ability that consciously applies logic from premises to conclusion” [2, 3]. As Garcez and Lamb said, “Reasoning can take place either symbolically or within the neural network via the statistical learning”[3].  In other words, if we ignore the interactions between humans and machines, the induction and logical reasoning of machines can be performed in either a distributed or symbolic fashion. Therefore, we think the reasoning ability learned from statistical learning is an important part of artificial intelligence.
>
> In the meanwhile, the symbolic form of logic and interpretability (such as the studies [4, 5]) are always preferred since they are easy to understand and validate for humans. Although our work follows the traditional literature of visual reasoning to make conclusions by neural networks, our method makes important contributions to explicitly learning robust rule representations and providing interpretability of the rule. In the future, we could further extend this work to a neural-symbolic method to address the visual reasoning task. We have discussed this point in our revision (Section 5).
>
> **Q3: A few neuro-symbolic methods are missing.**
>
> **A3:** We have added and discussed them in our revision to provide a complete picture of the abstract reasoning field (Section 2).
>
> [1] Sheng H., Yuqing M., Xianglong L., Yanlu W., and Shihao B.. Stratified rule-aware network for abstract visual reasoning. In AAAI, pages 1567–1574, 2021.\
> [2] Proudfoot, M., and Alan R. L. The Routledge dictionary of philosophy. Routledge, 2009.\
> [3] Artur S. G., Luis C. L., and Dov M G. Neural-symbolic cognitive reasoning. Springer Science & Business Media, 2008\
> [4] Chi Z., Baoxiong J., Song-Chun Z., Yixin Z. Abstract spatial-temporal reasoning via probabilistic abduction and execution. In CVPR, 2021.\
> [5] Chi Z., Sirui X., Baoxiong J., Yingnian W., Song-Chun Z., and Yixin Z. Learning algebraic representation for systematic generalization in abstract reasoning. arXiv preprint arXiv:2111.12990, 2021.

---

> > ### Comment · Reviewer_xDma · 2022-08-09
> > **Thanks for addressing my concerns**
> >
> > I have read the rebuttal and other reviews. I do not have additional questions and will raise my rating to 6.

---

> > > ### Author Response · Authors · 2022-08-09
> > > **Thanks for your support**
> > >
> > > Thanks again for your insightful reviews. We are grateful to see that our response addresses your concerns, and appreciate your positive feedback.
> > >
> > > We are glad to provide additional responses if you have any further questions.

---

### Official Review · Reviewer_4Dou · 2022-07-11

**Rating:** 8
**Confidence:** 5
**Soundness:** 4 excellent
**Presentation:** 4 excellent
**Contribution:** 4 excellent

**Summary:**

This paper proposes a novel method for visual abstract reasoning tasks, focused on the RPM-like benchmarks PGM and RAVEN. Some of the key features of the new method are the use of a learnable discrete representation to encode abstract rules, and a self-supervised objective that requires the model to use the encoded rule to fill in masked panels. The model achieves competitive results on both benchmarks, and shows particularly strong performance in the OOD regimes of the PGM dataset. There are also some interesting analyses of the rule representations learned by the model.

**Questions:**

I don't have any questions, other than regarding some of the missing details about hyperparameters and training.

**Limitations:**

I don't envision any potential negative societal impact from this work.

**Strengths And Weaknesses:**

## Strengths
- The model achieves excellent results, surpassing or competitive with the state-of-the-art on both benchmarks, and with particularly strong performance on the OOD regimes of PGM, which are arguably the most important part of these benchmarks.
- The evaluation is very thorough, evaluating the model on all formats and generalization regimes for the two major benchmarks in this area.
- The inductive biases introduced in this method are quite interesting. Using discrete encodings of the rules likely forces them to be more abstract, which presumably contributes to the OOD performance. The self-supervised objective ('internal inference') is an interesting way to capture the kind of 'hypothesis testing' that characterizes the way human reasoners solve these problems (internally generating proposals for the abstract rules, and then checking to see if they explain the presented panels).
- The ablations nicely demonstrate the contributions of these two components.
- The authors have included some code in the supplemental material, and plan to release all code upon acceptance.

## Weaknesses
- The primary weakness is that there are a number of important details missing that need to be included, perhaps in an appendix, e.g. training details (number of epochs, batch size, learning rate etc), model hyperparameters, and the computing machinery that was used to train the models. These should be included in the final version of the paper.

I also have a few other minor notes:
- I like the rule interpretability analysis, but it would be more informative if it were more comprehensive, rather than only focusing on rules applied to line color.
- It would be informative to also test the proposed method on this recently released variation on RAVEN: https://arxiv.org/abs/2206.14187
- It would be good to explicitly mention the connection between the 'internal inference' objective and self-supervised learning.
- The object encoding method employed here is very inflexible, relying on the rigid spatial structure of these problems. I think it's ok in the context of this work, but it would be good to mention this as a limitation and a potential avenue for future work.

---

> ### Author Response · Authors · 2022-08-02
> **Response to Reviewer 4Dou**
>
> We thank the reviewer for the instructive comments and the nice summary of our contributions. Below, we will make a point-to-point response to each specific comment.
>
> **Q1: Implementation details.**
>
> **A1:** We have provided more implementation details to ensure the reproduction of our results, including the hyperparameters, the training settings, and the computing machinery. Due to the page limitation, we have put this part to Appendix. Besides, we will release both the training and the test code upon acceptance.
>
> **Q2: More interpretability analysis.**
>
> **A2:** In addition to the [・,line, color] rules, we also visualize the selection frequencies of the quantized code for the [s, shape, color] rules, where $s \in S$. The distributions of the selection frequencies are shown in Appendix Fig 2. We notice that all the five rules share the same code E397 as their top-1 code, indicating that the code E397 is highly relevant to the [・,shape, color] rules. These results further validate the strong interpretability of the rule representations in ARII.
>
> **Q3: Test on a newly released dataset.**
>
> **A3:** We will test our model on this newly released dataset. Considering that this dataset was released after we submitted our paper, we need some time and effort to preprocess the data and train the model.
>
> **Q4: The connection between the 'internal inference' objective and self-supervised learning.**
>
> **A4:** The internal inference can be viewed as one type of self-supervised learning algorithm for the rule representation learning. But the traditional self-supervised learning models are usually trained first and finetuned for the downstream tasks afterward. Different from them, we explore an interesting strategy that performs the internal inference and the supervised learning simultaneously. We have added the above discussion in the revision (Appendix F).
>
> **Q5: Limitations on the object encoding method.**
>
> **A5:** We agree that the current object encoding is not inflexible and general. We will explore the object encoder that is more robust and flexible to represent the objects, such as using the object detector pretrained with natural images. In the revised paper, we have discussed the limitation of the current object encoder and the potential avenue for future work (Section 5).

---

> > ### Comment · Reviewer_4Dou · 2022-08-02
> > **Reply**
> >
> > Thanks to the authors for these responses. I am glad to hear there will be more implementation details in the revised version, and also discussion of the relation with self-supervised learning, and the limitation of the current object encoding method. For Q3 (the newly released dataset), I am indeed interested to see how the model performs, but I intentionally put this in the 'minor notes' section because I didn't want to emphasize it as something the authors absolutely need to do, given that the dataset was recently released. Got Q2 (interpretability analyses), it is good to include additional results for shape color rules, but I think it would be better still to do something more systematic. It seems like the key question here is whether the 'rule encodings' really capture the underlying rules in an explicit way. A simple way to determine this would be to see whether it's possible to train a (single-layer) classifier to classify the triples associated with each problem, given the rule encodings. That would tell us, in a more systematic manner, to what extent the rules are explicitly encoded in these embeddings across all rule types.

---

> > > ### Author Response · Authors · 2022-08-04
> > > **Response to Reviewer 4Dou**
> > >
> > > Thank you very much for your generous response and we appreciate your efforts in the review process. As suggested by the reviewer, we use a single-layer feedforward neural network to investigate whether the rule representations capture the underlying rules. In particular, we first collect the rule representations from all the instances from the test set on the PGM dataset. All the instances are not seen before by the model and are split into training, validation and test sets following a ratio of 8:1:1. Then, since there are 15 rules in the dataset, we train a linear layer (400 $\times$ 15) followed by the softmax function to classify the rule representations (400 dimensions). The test results are reported as follows.
> > > |      | Random Chance | SRAN | ARII w/o training | ARII  |
> > > |:---------------:|:------:|:--------------------------------:|:--------------------:|:--------------------:|
> > > | Classification Accuracy | 6.7% | 54.8% | 13.1%    | 74.1%  |
> > >
> > > We observe that the learned rule representations achieve 74.1% classification accuracy while the random chance only yields 6.7%. We also perform the same analysis on both the ARII model that was not trained (i.e., the weights of ARII were randomly initialized) as well as one baseline model (i.e., SRAN [1])  for comparison. We notice that the rule representations of randomly initialized ARII obtain an accuracy of 13.1%, indicating that ARII has learned the general rules underlying the specific instances in an explicit way. Besides, the representations produced by SRAN merely yield 54.8%, largely lower than those of ARII. The above systematic comparisons demonstrate that ARII has an impressive capacity for learning rules compared to the previous methods.
> > >
> > > We found this suggestion is very helpful in improving the quality of our submission. We will incorporate our response into our final version.
> > >
> > > [1] Sheng H., Yuqing M., Xianglong L., Yanlu W., and Shihao B.. Stratified rule-aware network for abstract visual reasoning. In AAAI, pages 1567–1574, 2021.

---

> > > > ### Comment · Reviewer_4Dou · 2022-08-08
> > > > **Followup**
> > > >
> > > > Thanks very much to the authors for this additional analysis. This is exactly the sort of thing that I had in mind, and the comparison with SRAN and also to the untrained version of the current model are very informative. I think this much more clearly establishes that the current model is, to a significant extent, learning explicit representations of the underlying rules.

---

> > > > > ### Author Response · Authors · 2022-08-09
> > > > > **We are glad to have more discussions with you**
> > > > >
> > > > > Once again, we are grateful for your efforts and support in the review process, in particular for further responding to our updated results in the rebuttal.
> > > > >
> > > > > In addition, please feel free to raise any additional questions/concerns about our work. We would provide you with additional responses and promise to include most of these clarifications in the final revised version to make our paper clear.

---

### Official Review · Reviewer_ky2S · 2022-07-11

**Rating:** 7
**Confidence:** 5
**Soundness:** 4 excellent
**Presentation:** 4 excellent
**Contribution:** 3 good

**Summary:**

The authors propose a novel framework Abstract Reasoning via Internal Inferences (ARII) for the task of solving abstract visual reasoning problems. In particular, the ARII model is designed to solve the task of selecting the correct candidate panel for a Raven's Progressive Matrices (RPM) reasoning problem. The proposed method consists of three components:

1. A rule encoder network that takes the first two context rows of the RPM matrix, and generates vector quantized codes representing the abstract rule encoded in the images. The encoder works by generating CNN embeddings for each panel, which are then split into groups and concatenated based on their position. These image encodings are once again split into positions for each of the six image panels and then re-combined together to generate a continuous rule representation. The continuous rule representation is then discretized into the Ke X D dimensional code book, which is trained together with the encoder.
2. A reasoner which takes each candidate panel as a potential rule with the context panels in the third row, calculates the rule encoding of this candidate rule with rows 1 and 2, and then selects the candidate rule (panel) which has the highest similarity with the rule encoded by rows 1 and 2.
3. A generative internal inferrer, that masks out each image sequentially out of the six context images in the first two rows, and aims to generate the original rule encoding with one context image missing by minimizing the distance between the original rule and the rules extracted with a panel missing.

ARII outperforms existing methods when it comes to generalizing to novel test domains on the i-RAVEN and PGM datasets. Ablation studies performed by the authors demonstrate the utility of both the discretized code book as well as the internal inference mechanism.


**Questions:**

Could the authors further validate their claims that the codes learned by ARII are instance-invariant? In abstract reasoning terms this would involve that the codes learned for one particular rule (such as AND) will be similar irrespective of the visual domain. The ablation studies done by the authors do not demonstrate this. Please see the limitations highlighted in the review for one possible suggestion to demonstrate this.

**Post-Rebuttal Edit:** See response to author's rebuttal below. Score increased to a 7 post-rebuttal.

**Limitations:**

The authors did a good job in explaining several assumptions in their modeling process, such as space of rules is not large. Several of the ideas utilized in the ARII model build upon previous research (such as splitting and grouping panel embeddings like the Scattering Composition Learner, row wise embeddings relying on feature locations like Multiplex Graph Network, etc.) While many of these papers are referenced, the authors fail to draw connections and provide reasonable credit to the authors of these works in their Related Work section. It would be great if the authors can highlight these connections, as it will help future research in abstract visual reasoning and would be only fair to the authors of previous works.

**Strengths And Weaknesses:**

Strength:

1. The paper is extremely well written: the RPM problem is presented succinctly, the ARII model is explained clearly and each component is motivated well, and the training procedure, results and ablation studies are easy to follow.

2. The idea of utilizing a learnable vector quantized codebook for representing abstract rules in PGM problems is quiet novel, and works well in practice as seen from the generalization results of ARII.

3. Under the scenario where auxiliary information such as symbolic rule embeddings are not used for training, it demonstrates state-of-the-art performance on four generalization splits of the i-RAVEN dataset, and five generalization splits of the PGM dataset.

Weakness:

1. In equation 1, the authors make an implicit assumption that the RPM rule is encoded row-wise in the image panels. While this holds true for the i-RAVEN dataset, it is not necessarily true for PGM. The authors work around this limitation by adding the transposed PGM matrices for training. Several other models compared in the paper do not explicitly rely on this assumption, and hence it would only be fair to compare these models with this extra training step included.

2. In the ablation study, the authors show that the codes E-358 and E-181 occur in the top three codes for each possible relation in the visual domain of [., line, color] rules. This appears counter-productive to the argument that the ARII model has learned instance-invariant representations, since if it were indeed instance-invariant then the visual domain should have little to no impact on the rule representation. A better way to demonstrate this would have been to take the codes for [AND, ., .] across multiple visual domains and show that the codes are consistent with various object and attribute values.

---

> ### Author Response · Authors · 2022-08-02
> **Response to Reviewer ky2S**
>
> We thank the reviewer for the nice summary of our contributions. Below, we will make a point-to-point response to each specific comment.
>
> **Q1: Transposed PGM matrices for training.**
>
> **A1:** We conduct an ablation study to see the influence of the transposed PGM matrices on the reasoning performance. In particular, we select two competing baselines whose codes were released (i.e., SCL [1] and SRAN [2]). We train these baselines with the PGM dataset and its transposed PGM matrices. The results are reported in the below table.
> |            | SRAN  |      SRAN           | SCL               |    SCL             |
> |:-------|:------|:------|:------|:-----|
> | Regime        | w/o transposition | w transposition | w/o transposition | w transposition |
> | Interpolation | 44.9              | 47.8            | 67.1              | 63.6            |
> | H.O.A.P        | 30.9              | 31.6            | 32.5              | 32.1            |
>
> We observe that SRAN trained with the additional dataset only has marginal improvement. The performance of the SCL model trained with the additional transposed dataset even presents decreases. These results show that the benefit of the transposed matrices to the baselines is limited. This is because most of the previous visual reasoning methods (e.g., SRAN and SCL) could extract both row-wise and column-wise features. Therefore, the transposed PGM matrices could hardly provide more information than the original matrices for the baseline models. Therefore, our comparisons are fair. We have added these results in the revision (Section 4.4, Appendix E)
>
> **Q2: Are representations of rules Instance-invariant? This appears counter-productive to the argument that the ARII model has learned instance-invariant representations, since if it were indeed instance-invariant then the visual domain should have little to no impact on the rule representation.**
>
> **A2:** The abstract rule in visual reasoning IS HIGHLY relevant to the visual domains. As defined in the original paper on the PGM reasoning task, the abstract rule in the PGM matrix is a set of triples [r; o; a] (r: relation, o: object, a: attribute), which stands for that, in a row of images, the attribute a of the object o has a relation of r. Therefore, a rule in the abstract reasoning is still respective to the visual domain. In the meanwhile, the above rule is still abstract, because the rule is invariant to the positions, directions, and other attributes of the object.
>
> In addition, Figure 3 shows that the rule representations are clustered according to individual rule triples and ignore the identity of the specific instances, clearly demonstrating the rule representations are instance-invariant.
>
> **Q3: Showing that one particular rule (such as AND) will be similar irrespective of the visual domain.**
>
> **A3:** “AND” is not a particular rule in the abstract reasoning tasks. The abstract rule in the PGM matrix is a set of triples [r; o; a] (r: relation, o: object, a: attribute). “AND” is just one type of relation in the rule. Actually, the “AND” relation in different rules (denoted by [AND, .,.]) presents dramatically distinct properties. For example, the “AND” relation of the shape of lines is quite different from the “AND” relation of the color of lines. Therefore, analyzing the “AND” relation is infeasible.
>
> **Q4: Draw connections to the previous related works.**
>
> **A4:** In the revised submission, we have updated the related work and the methodology section to provide reasonable credit to the works that the ARII method is built upon.
>
> [1] Yuhuai W., Honghua D., Roger G., and Jimmy B.. The scattering compositional learner: Discovering objects, attributes, relationships in analogical reasoning. arXiv preprint arXiv:2007.04212, 2020\
> [2] Sheng H., Yuqing M., Xianglong L., Yanlu W., and Shihao B.. Stratified rule-aware network for abstract visual reasoning. In AAAI, pages 1567–1574, 2021.

---

> > ### Comment · Reviewer_ky2S · 2022-08-08
> > **Response to author's rebuttal**
> >
> > Thanks a lot for taking the time to run the suggested ablations, as well as expanding upon the related work section of your paper. I believe both of these increase the quality of the paper, as well as its overall readability. Hence, I have increased my score to a 7.
> >
> > Regarding the invariance wrt rules, I understand that the relation is only a part of the rule triplet. Perhaps there was some miscommunication in what the authors meant and what I inferred: the authors meant that the representations learned by ARII are invariant of the training instance (i.e. exact ordinal values of the attributes) whereas I inferred it as claiming that they are invariant to the visual domain (i.e. both the value and identity of attributes and objects). I concede that the authors claims apear correct based on their interpretation of instance-invariance, but also note that they are not instance-invariant in the much more general sense that I inferred (or other readers might infer).
> >
> > While "The abstract rule in visual reasoning IS HIGHLY relevant to the visual domains" is true, the problem statement in Raven's Matrices focuses on learning representations that are invariant to visual domains. For a concrete example, let us consider the generalization splits the PGM dataset includes: novel attribute values (interpolation, extrapolation), held-out object-attribute combination (line-type, shape-color), and held-out attribute pairs. These are all generalization splits where the model needs to learn rule representations that can transfer to the test set despite being presented with a new visual domain. The generalization requirement for the test splits in PGM focuses on generalizing to new visual domains (i.e. object and/or attribute combination) not on generalizing to new relations. This is based on cognitive theories of humans (e.g. the structure mapping theory, where generalization to new analogies focuses on generalizing the relation across domains instead of generalizing the object and attributes.) Hence, I suggested the authors run an ablation study visualizing the code book for individual relations.

---

> > > ### Author Response · Authors · 2022-08-09
> > > **Response to Reviewer ky2S**
> > >
> > > Thanks for your insightful and positive feedback.
> > >
> > > Since relations are much more abstract and general than the rule triples, it is more complicated to learn the relation itself. And the relation visualization results are not very significant (see Appendix G for specific analysis).
> > >
> > > To further investigate whether our method has learned the underlying relations, we conduct another experiment to classify the relations. In particular, we use a single-layer classifier to classify the relations based on the rule representations associated with each problem. We first collect the rule representations from all the instances from the test set on the PGM dataset. All the instances are not seen before by the model and are split into training, validation and test sets following a ratio of 8:1:1. Then, since there are 5 relations (AND, OR, XOR, progression, and consistent union) in the dataset, we train a linear layer (400$\times$5) followed by the softmax function to classify the rule representations (400 dimensions). The test results are reported as follows.
> > >
> > > |      | Random Chance | ARII w/o training | SRAN | ARII |
> > > |:--------------:|:------:|:--------------------------------:|:--------------------:|:--------------------:|
> > > | Classification Accuracy | 20% | 21.8%    | 59.4% | 72.1% |
> > >
> > > It can be seen that the single-layer classifier can well classify the relations based on the rule representations extracted from ARII (72.1% classification accuracy) associated with each problem. The random chance and the result from ARII that was not trained (i.e., the weights of ARII were randomly initialized) are 20% and 21.8%, respectively. We also compare our model with a baseline model, SRAN [1], which yields 59.4% classification accuracy, largely lower than that of ARII. The comparison results demonstrate that ARII can capture the underlying relations in different visual domains, and therefore learn the invariant representations to some extent.
> > >
> > > We found that this suggestion is very helpful in improving the quality of our submission. We will incorporate our response into our final version.
> > >
> > > [1] Sheng H., Yuqing M., Xianglong L., Yanlu W., and Shihao B. Stratified rule-aware network for abstract visual reasoning. In AAAI, pages 1567–1574, 2021.

---

### Official Review · Reviewer_bvvT · 2022-07-11

**Rating:** 6
**Confidence:** 4
**Soundness:** 3 good
**Presentation:** 3 good
**Contribution:** 3 good

**Summary:**

The paper addresses the important issue of abstract reasoning, in particular the ability of neural networks at performing extrapolation.

The paper's main goal is summarised by the authors as to "ameliorate the reasoning generalizability by learning a robust rule representation". This needs to be reconciled with the goal of addressing extrapolation, since generalizability is something else.

The authors claim to learn rule representations and to perform reasoning. Many ML systems learn rules. It is important to note that the rule representations are embeddings and that the reasoning is by similarity only. Looking at the paper from that perspective makes the claim made around visualization and interpretation of the rules seem rather vague.

Having said that, the task at hand being considered in the paper is very interesting from a reasoning point of view. And the improved results obtained are valid and relevant and should not be ignored.

A main claim of the paper is that other approaches may "require additional prior information, which is not a general way to address the reasoning problem".

**Questions:**

It is not easy to see the relationship between Fig 1 and the rule encoder, the reasoner, and the internal inferrer. Is this done on purpose?

A main claim of the paper is that other approaches may "require additional prior information, which is not a general way to address the reasoning problem". Other approaches may provide a general mechanism for reasoning. When it comes to a fair comparison, how can you be sure that the "internal inference" process isn't a way of providing additional information?

The "internal inference" process seems to function in such a way as to turn propositional rules into rules with variables. Is this a fair description of the purpose of the proposed process? You state that "we randomly mask one panel of the first two rows and ask the referrer to infer the masked panel based on the rule representation. We perform the internal inference for multiple times to make the rule representation invariant to specific instances." To answer my question above, this process needs to be specified well and analysed from the perspective of reasoning capabilities.

How do we know that the results derive from better reasoning rather than e.g. the combinatorial benefit of using r1&2 with r1&3 and r2&3? Did the ablation study conclude anything about this? How do we check whether this internal inference is indeed producing the expected "more abstract" rules (with variables)?

When one says that one evaluates "reasoning ability", one would wish to evaluate what kinds of reasoning take place, not simply evalute improvements in test set accuracy. Have you evaluated what happens in a situation where the reasoning required to obtain the correct answer is generally seen as harder than the normal type of reasoning?

**Strengths And Weaknesses:**

The paper is well-written and well-organized, although the captions of the figures could have been more informative, e.g. Figs 3 and 4 are difficult to grasp.

The task at hand being considered in the paper is very interesting from a reasoning point of view. And the improved results obtained are valid and relevant.

The related work section is generally limited to work on the specific datasets, e.g. RAVEN and variations thereof.

A number of acronyms is used in the paper without stating first what they stand for, sometimes even without providing a reference, e.g. PGM.

Robustness should also be defined (ideally). The most problematic part of the paper is the lack of a definition of "reasoning". Reasoning in neural networks is an important endeavour. It is nowadays common for neural net papers to refer to reasoning without providing any definition of what kind of reasoning or language that is being considered (and there are many). These are all formally well-defined in the area of knowledge representation in AI. The area of neurosymbolic AI has taken those definitions on board. If the field is to develop as needed in this direction then it is important to consider the work that goes on since the 1990's in knowledge representation and neurosymbolic AI. I'd refer the authors at least to:

https://link.springer.com/book/10.1007/978-3-540-73246-4
https://ebooks.iospress.nl/volume/neuro-symbolic-artificial-intelligence-the-state-of-the-art

And specifically on adding knowledge to neural nets:
https://www.ijcai.org/proceedings/2017/221
https://www.researchgate.net/publication/283897473_Semantic-based_regularization_for_learning_and_inference
https://arxiv.org/abs/1605.06523

And for a recent survey:
https://arxiv.org/abs/2012.05876

---

> ### Author Response · Authors · 2022-08-02
> **Response to Reviewer bvvT (Part 1/2)**
>
> We thank the reviewer for the instructive comments and are grateful for the time you spent with our work. We are also glad for the acknowledgment that the problem we are working on is interesting.
>
> **Q1: Captions of the figures could have been more informative.**
>
> **A1:** We have added more descriptions to the captions of the figures to make them easier to understand in the revised manuscript.
>
> **Q2: Related work.**
>
> **A2:** We have extended the related work section to better situate our work on both the abstract reasoning and the neural-symbolic literature.
>
> **Q3: Acronyms are used in the paper without stating first.**
>
> **A3:** We have modified it in the revision.
>
> **Q4: Robustness should be defined.**
>
> **A4:** The robustness of a machine learning model is defined as: the closeness between the testing error to the training error when the testing samples are “similar” to training samples [1]. Figure 3 in the main text shows that the rule representations in ARII are mainly dependent on the rule identity and irrespective of the specific samples (including noises). Therefore, the learned rule representations in ARII are robust. We have added the definition of robustness in the revised version (Section 3.5).
>
> **Q5: Lack of a definition of "reasoning".**
>
> **A5:** One of the widely used definitions is “reasoning is the ability that consciously applies logic from premises to conclusion” [2,4]. For neural networks, they use different layers of the distributed neurons to implicitly represent the premises, the logic and the temporal results, and formulate the final conclusion-making process as the classification or generation task. Although it is difficult to analyze the reasoning process of neural networks, the reasoning performance has been significantly improved by neural networks [3,5,6,7,8]. As the reviewer pointed out, it is a promising way to combine the symbolism and the connectionism to build an interpretable, robust, and accurate reasoner in future work. We have added the above discussion in the revision (Section 5).
>
> **Q6: It is not easy to see the relationship between Fig 1 and the rule encoder, the reasoner, and the internal inferrer.**
>
> **A6:** Fig 1 is to illustrate the overall motivation of this paper, that is, leveraging internal inference to build a robust rule representation for visual reasoning. In order to highlight the key points, we have omitted the particular modules (such as rule encoder and reasoner). Readers can easily access the definition and description of these modules in Fig 2 and the main text.
>
> **Q7: How can you be sure that the "internal inference" process isn't a way of providing additional information?**
>
> **A7:** The standard to justify whether one model has accessed the additional information is simple, that is, whether the input and the supervision signals are the same with the baselines. Some approaches utilized extra labels, such as the meta-target, which encodes the rule in the instance [9, 10]. They train the model not only to predict the answer but also the meta-target. That is, they use additional supervision signals to enhance the model performance. By contrast, our method only takes the incomplete matrix as input and the missing image as the supervision signal, without any additional information for the training or testing.
>
> **Q8: The "internal inference" process seems to function in such a way as to turn propositional rules into rules with variables. Is this a fair description of the purpose of the proposed process?**
>
> **A8:** The internal inference process does not turn propositional rules into rules with variables. Reviewer 4Dou has a nice summary of the internal inference process, i.e., it is a “kind of 'hypothesis testing' that characterizes the way human reasoners solve these problems (internally generating proposals for the abstract rules, and then checking to see if they explain the presented panels)”. In order to enhance the above process and obtain the robust rule representations, we perform the internal inference for multiple times by randomly masking the different images in the given two rows (i.e., the rows that the internal inferrer used). Also, the ablation study (Table 3) validates our design, that is, the internal inference could significantly improve the robustness of the reasoning.

---

> > ### Author Response · Authors · 2022-08-02
> > **Response to Reviewer bvvT (Part 2/2)**
> >
> > **Q9: The combinatorial benefit of using r1&2 with r1&3 and r2&3.**
> >
> > **A9:** Following the reviewer’s suggestions, we conduct an additional study on the combinatorial benefit of using r1&2 with r1&3 and r2&3. In particular, we remove the information of r2&3 from the inputs of the reasoner and report the performance in the below table. We observe that the reasoning performance decreased slightly compared to the original model. This result indicates that the combination of using r1&2 with r1&3 and r2&3 is beneficial to the reasoner but it is not the main reason for the performance gain of ARII. Instead, Table 3 (the third column) in the main text presents that the internal inference could significantly improve the robustness of the reasoning. We have added these results in our revision (Appendix C).
> > | Regime        | ARII (r1&3 & r2&3) | ARII (w/o r2&3) |
> > |---------------|------|--------------------------------|
> > | Interpolation | 71.6 | 69.5              |
> > | H.O.A.P       | 61.6 | 57.1                |
> >
> >
> > **Q10: Have you evaluated what happens in a situation where the reasoning required to obtain the correct answer is generally seen as harder than the normal type of reasoning?**
> >
> > **A10:** The PGM dataset, especially the generalization regime, is well-suited for harder reasoning. The rule in the PGM dataset is represented as a set of triples [r; o; a] (r: relation, o: object, a: attribute) and there are 29 possible unique triples. For example, in the held-out triple regime, seven of these triples are held out for the test set. The held-out triples never occurred in questions in the training set, and the problems in the test set contained at least one of them. In traditional reasoning tasks, the test set generally contains the same rules as the training set, just with the test image changed. Therefore, the generalization regime in PGM is a harder way to test the reasoning ability since the rule distributions of the training and test sets are different.
> >
> >
> > [1] Xu, H., and Mannor, S. (2012). Robustness and generalization. Machine learning, 86(3), 391-423.\
> > [2] Proudfoot, M., and Alan R. L. The Routledge dictionary of philosophy. Routledge, 2009.\
> > [3] Donadello, I., Serafini, L., and Garcez, A. D. A. Logic tensor networks for semantic image interpretation. IJCAI, 2017.\
> > [4] Artur S. G., Luis C. L., and Dov M G. Neural-symbolic cognitive reasoning. Springer Science & Business Media, 2008.\
> > [5] Hitzler, P. Neuro-Symbolic Artificial Intelligence: The State of the Art. 2021.\
> > [6] Garcez, A. D. A., and Lamb, L. C. Neurosymbolic AI: the 3rd wave. arXiv preprint arXiv:2012.05876. 2020.\
> > [7] Diligenti, M., Gori, M., and Sacca, C. Semantic-based regularization for learning and inference. Artificial Intelligence, 244, 143-165. 2017.\
> > [8] Zheng, Z., Wang, W., Qi, S., and Zhu, S. C. Reasoning visual dialogs with structural and partial observations. In CVPR. 2019.\
> > [9] Mikołaj M. and Jacek M. Multi-label contrastive learning for abstract visual reasoning. arXiv preprint arXiv:2012.01944, 2020.\
> > [10] Duo W., Mateja J., and Pietro L. Abstract diagrammatic reasoning with multiplex graph networks. In ICLR, 2020.

---

### Official Review · Reviewer_fEdf · 2022-07-11

**Rating:** 7
**Confidence:** 3
**Soundness:** 3 good
**Presentation:** 3 good
**Contribution:** 3 good

**Summary:**

The authors propose ARII, a new framework to solve the RPM datasets. The main idea is to try to learn a reusable rule representation from an instance of the task, such that it can be applied to other tasks when applicable. They achieve this by mapping individually learned rule vectors to a discrete representation using vector quantization and then reusing those rules to infer some of the already present panels in the RPM instance. Overall, they show that ARII is able to perform better, specifically on larger grids and more difficult settings.

**Questions:**

Writing suggestions:
* It would help to have some background on the details of the vector quantization steps involved in the discretization of the rule embeddings. Maybe a motivating figure to explain with an example what q_l, etc., mean can make that section more accessible.

**Limitations:**

Please add a section to describe the limitations of the work. I believe the major limitation is deciding the size of the vector quantization you do to obtain unique rules. Potentially, there can be new rules for each instance of the dataset, where your basic hypothesis breaks. Try to add some discussion along these or other relevant points.

**Strengths And Weaknesses:**

Strengths:
* The paper is nicely motivated in general. The design of each module is well described and easy to follow through
* Final results on both performance and interoperability are quite impressive

Weakness:
* The motivation for the design choices is not well supported in the ablation. I'm specifically not convinced about the design choices of the internal inferences (Section 3.5). Why does the process need to be generative instead of classification? For negative answer choices, you can fall back on the existing options in the instance. The correct answer is the particular image you mask out. The advantage of this is you can potentially reuse the reasoner module for this part as well. This can at least be an additional idea to try for ablations, if not the main approach.

---

> ### Author Response · Authors · 2022-08-02
> **Response to Reviewer fEdf**
>
> We thank you for the review and are grateful for the time you spent with our work.
>
> **Q1: Design choices of the internal inference.**
>
> **A1:** We have conducted an ablation study on the internal inference to investigate the role played by the generative process. In particular, we replace the generative process with classification. In the classification task, the input is the two rows of images where one image is masked with blank (i.e., $I_m$ in Equation 14, the same as the original ARII). The classification choices include the existing options in the instance and the particular image we mask out. The inferrer reuses the reasoner module to make the classification decisions. Apart from the above changes, the other settings are the same as the original ARII.
>
> The following table reports the results of the classification variant of ARII on the interpolation regime of the PGM datasets. We observed that the classification variant yields slightly lower performance than the generative variant on the interpolation regime. These results indicate that the classification process could also serve as a reasonable task in the internal inference module of ARII but the generative process is better. This ablation study further demonstrates that the internal inferences play a critical role in visual reasoning no matter which inference tasks are performed. We have added the above results in the revision (Appendix B).
>
> | Regime        | ARII (generation) | ARII (classification) |
> |---------------|------|--------------------------------|
> | Interpolation | 71.6 | 62.4              |
> | H.O.A.P       | 61.6 | 59.4                |
>
> **Q2: The details of the vector quantization.**
>
> **A2:** We have added more details on vector quantization in the revision (Section 3.3) and provided a figure (Appendix Fig 1) to explain the computation process.
>
> **Q3: The size of the vector quantization.**
>
> **A3:** Since our method selects multiple quantized vectors from the codebook to compose the final rule representation, the combinational number of rules could be huge (i.e., $A_{K_e}^{K_r}$, $K_e$=512, $K_r$=80). Therefore, ARII has sufficient expressive capacity in the vector quantization to represent the visual rules. Following the comments of the reviewer, one of the limitations of ARII is performing reasoning on the case where the rule is brand-new and could not be represented with the previously learned patterns. However, this drawback is a general challenge for all the current methods.

---

### Meta-Review · Area_Chair_GxX9 · 2022-08-25

**Recommendation:** Accept
**Confidence:** Certain

**Metareview:**

I thank the authors for their submission and active participation in the discussions. The paper presents a method for rule representation learning that can be transferred accross tasks. All reviewers unanimously agree that this paper's strengths outweigh its weaknesses. In particular, reviewers found the method to be well motivated [fEdf], general [fEdf], novel [ky2S], tackling an interesting task [bvvT], achieving strong empirical results [4Dou] and the paper to be well written [bvvT,ky2S,xDma]. Thus, I am recommending acceptance of the paper and encourage the authors to further improve their paper based on the reviewer feedback.

**Award:**

No

---

### Decision · Program_Chairs · 2022-09-14

Accept